# Latitude and longitude as drivers of COVID-19 waves' behavior in Europe: A time-space perspective of the pandemic

**Alejandro Martínez-Portillo**[1]☺*, **David Garcia-Garcia**[2,3‡], **Inmaculada Leon**[2,3‡], **Rebeca Ramis-Prieto**[2,3‡], **Diana Gómez-Barroso**[2,3☺]

**1** Department of Epidemiology, Regional Health Council, Murcia, Spain, **2** Centro Nacional de Epidemiología, Carlos III Health Institute, Madrid, Spain, **3** Consorcio de Investigación Biomédica en Red de Epidemiología y Salud Pública (CIBERESP), Madrid, Spain

☺ These authors contributed equally to this work.
‡ DGG, IL, RRP and DGB also contributed equally to this work.
* amp.droit12@gmail.com

## Abstract

**Data Availability Statement:** Data are available in Our World in Data webpage: https://

### Background

Social restrictions and vaccination seem to have shaped the pandemic development in Europe, but the influence of geographical position is still debated. This study aims to verify whether the pandemic spread through Europe following a particular direction, during the period between the start of the pandemic and November 2021. The existence of a spatial gradient for epidemic intensity is also hypothesized.

### Methods

Daily COVID-19 epidemiological data were extracted from *Our World in Data* COVID-19 database, which also included vaccination and non-pharmacological interventions data. Latitude and longitude of each country's centroid were used as geographic variables. Epidemic periods were delimited from epidemic surge data. Multivariable linear and Cox's regression models were performed for each epidemic period to test if geographical variables influenced surge dates. Generalized additive models (GAM) were used to test the spatial gradient hypothesis with three epidemic intensity measures.

### Results

Linear models suggest a possible west-east shift in the first epidemic period and features a significant association of NPIs with epidemic surge delay. Neither latitude nor longitude had significant associations with epidemic surge timing in both second and third periods. Latitude displays strong negative associations with all epidemic intensity measures in GAM models. Vaccination was also negatively associated with intensity.

ourworldindata.org/coronavirus#explore-the-global-situation

**Funding:** The author(s) received no specific funding for this work.

**Competing interests:** The authors have declared that no competing interests exist.

## Conclusions

A longitudinal spread of the pandemic in Europe seems plausible, particularly concerning the first wave. However, a recurrent trend was not observed. Southern Europe countries may have experienced increased transmissibility and incidence, despite climatic conditions apparently unfavourable to the virus.

## Introduction

The SARS-CoV2 pandemic has resulted in more than 62 million cases and more than one million deaths in European countries as of November 23, 2021 (Our World in Data). The evolution of the pandemic has been characterized by a series of periods of epidemic growth–the so-called epidemic waves–of varying intensity and duration, the most intense and concentrated in time being those most feared by the health authorities, due to the consequences for healthcare and public health services.

As a result, throughout the pandemic, European countries have implemented a series of public health measures aimed at curbing transmission. Many countries initially took the decision to introduce lockdowns; however, after the initial period of confinement, the level of restrictive measures in the following months has been uneven, as have the cumulative incidence figures for each moment [1]. Vaccination campaign progress had seemed to have led to the definitive abandonment of certain restrictions, but the events of autumn 2021 surprised many European countries–who found their proportion of vaccinated population to be insufficient; there were consequent increases in incidence figures, hospitalizations and ICU admissions [2].

The severity of social restrictions and the immunity levels achieved through vaccination are well-established factors [3, 4] that can, to a large extent, explain the behaviour of the pandemic in Europe–both in its spread across the continent and in the intensity found in each location. However, the influence of geography on the epidemic's behaviour has been widely debated in the scientific literature, with disagreements as to which variables are most important in the European region [5–7].

However, to our knowledge, no articles have explored these issues across the full extent of the pandemic and the whole continent yet. Therefore, based on the hypothesis that both the dispersion of COVID-19 in Europe and the intensity with which it manifests may be motivated by environmental factors linked to the countries' geographical positions, this study had two objectives for the period from the beginning of the pandemic to November 2021 and for each epidemic period: first, test the hypothesis that the pandemic moves through Europe in a particular direction and, second, to verify the existence of a spatial gradient for epidemic intensity.

## Methods

### Data collection

43 countries were included in this study (Albania, Andorra, Austria, Belarus, Belgium, Bosnia and Herzegovina, Bulgaria, Croatia, Cyprus, Czechia, Denmark, Estonia, Finland, France, Germany, Greece, Hungary, Iceland, Ireland, Italy, Latvia, Liechtenstein, Lithuania, Luxembourg, Malta, Moldova, Monaco, Montenegro, Netherlands, North Macedonia, Norway, Poland, Portugal, Romania, San Marino, Serbia, Slovakia, Slovenia, Spain, Sweden, Switzerland, Ukraine and United Kingdom). The daily Covid-19 epidemiological data for each country (Fig 1)

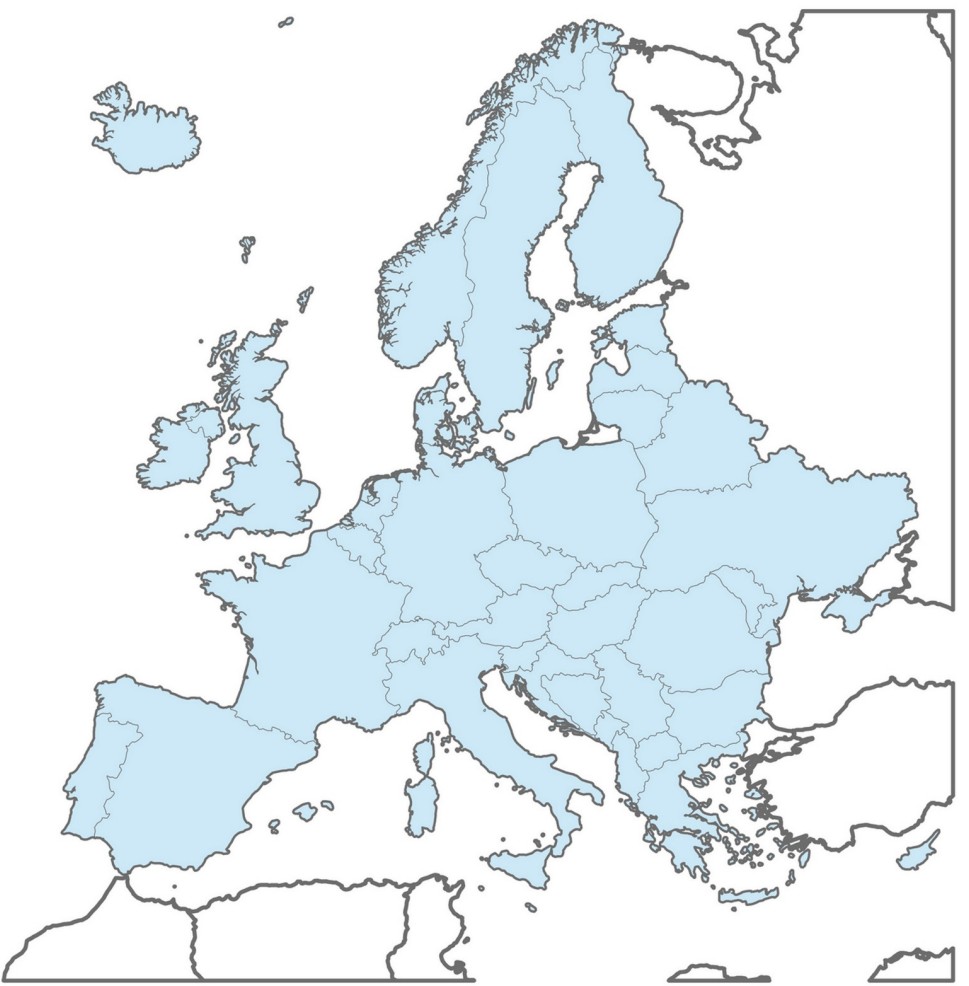

**Fig 1. In blue, countries included in the study.**

comes from the database collected by *Our World in Data* (https://github.com/owid/covid-19-data/tree/master/public/data). This information includes 14-day cumulative incidence (CI), new cases per million inhabitants and effective reproductive number (Rt) for each country. It also includes vaccination data for each country, collected as a percentage of people vaccinated with a complete course out of the total population, and on the number of daily diagnostic tests carried out nationally.

The stringency index for non-pharmacological interventions (NPIs) used was developed by the University of Oxford: the *Oxford Covid-19 Government Response Tracker* (OxCGRT) (https://github.com/OxCGRT/covid-policy-tracker). This is a tracker for monitoring the measures applied in various countries across the world; it calculates indexes with a range of 0 to 100 points, resulting from the average of the scores in a series of items evaluated for each country. The index used in this study is the so-called *Stringency Index*, which combines limitations on movement and on the number of people per meeting.

For the geographical position variables, the latitude and longitude of the centroid was calculated for each country from data obtained from the Geographic Information System of the European Commission (GISCO, Eurostat).

## Definition of epidemic surges and periods

To study the chronology of the pandemic, two different concepts have been defined: epidemic period and epidemic surge. Epidemic period alludes to the times in which there is an increase in the number of cases at the international level, while epidemic surge refers to these periods at a national level.

To evaluate the points at which the epidemic surged, we decided to use a definition provided by Zhang et al. al [8], according to whom there is an upward period of an epidemic wave when the effective reproduction number (Rt) stays above 1 for 14 or more consecutive days. To avoid confusion, we have renamed this "upward period" as an "epidemic surge". From this definition, the dates for which a period in which Rt fulfilled this condition were found for each of the countries studied. For each country the order in which these dates occurred was established. We grouped epidemic surges in temporal intervals, choosing the total number of temporal intervals to be the maximum number of surge dates observed at a single country.

Rigorous definitions of epidemic periods in Europe were not available, given the complexity of a precise delimitation of such a large-scale phenomenon. We selected three temporal landmarks in the pandemic development, which differentiated three stages with its own behaviour: an initial phase until the end of the initial lockdowns in the first months of 2020, a pre-vaccination phase until the beginning of 2021, and a last phase of widespread vaccination in Europe through 2021. Although most European countries went through all three of those stages, they did not do it simultaneously. We used the information provided by grouped surge dates to choose concrete, mutually-exclusive reference dates that fitted our landmarks, trying to keep every surge date where we considered it belonged while reducing overlapping to the minimum. These definitions were therefore non-formal and arbitrary, though based in data and epidemiological knowledge. Once these limits had been established and all surge dates had been reclassified as belonging to the first, second or third periods, in the cases where a country had several epidemic surges attributed to the same period, the first surges were selected in chronological order, so that each country had a single epidemic surge date per period.

## Analysis

**Regression models for epidemic surge.** To investigate the association between geographical location and the moment of epidemic surge in each period, a multivariable linear regression was used that took latitude and longitude as independent variables, as well as the proportion per thousand people of tests performed and the stringency index. The latter were introduced into the model as the average of those variables in the 14 days prior to the surge dates for each country. For vaccination percentage, the number of people fully vaccinated was taken at 14 days before the surge date, assuming that this would correspond to the percentage of people with maximum immunity at the moment of epidemic surge. As a dependent variable, the date of epidemic surge was used, having converted the dates selected for each country to the ordered position those days occupy in the study period, starting with the year 2020 and ending on 11/23/2021.

Since the only available precedent for this type of analysis was performed using linear regression [6], we also carried out a survival-type analysis with Cox regression to reassess the results of the linear regression, as a test for robustness. For this analysis, the time elapsed until epidemic surge in each country acted as a dependent variable, and the independent variables were: the latitude, longitude, the OxCGRT and the percentage of people fully vaccinated in the total population, the latter two introduced as in the linear regression. Note that in this model, values below 1 for the OR of a given variable correspond to delayed surge dates associated to the increase of the variable.

**Epidemic intensity models.**   On the other hand, to estimate the effect of the variables on the transmissibility of SARS-CoV2 and on epidemic intensity, generalized additive models (GAM) were used in which 14-day accumulated incidence, new cases per million and Rt were used as response variables in three different models. NPIs, the vaccination percentage, the number of tests per million, latitude and longitude, the moment in time (date, introduced as a non-linear variable) and, as random effects, the country and country-specific effects of NPIs and vaccination were introduced into the model as covariates.

Since the effect on the outcome variable of some of the factors studied is not immediate, a temporal delay has been introduced in the stringency index and vaccination percentage. This temporal delay has been set at 14 days.

All analyses have been carried out, and graphical elements produced, using the statistical software R, version 4.0.5 (analysis packages used: *survival*, *mgcv*).

## Results

### Epidemic surges and periods

The maximum number of surge dates observed at a single country was eight (Denmark, Moldova, Malta and Netherlands). Surge dates were therefore grouped into eight groups. Fig 2 shows the temporal distribution of epidemic surge dates grouped by their order of occurrence, in which the median, interquartile range (in blue) and 5th/95th percentiles (in grey) are indicated, showing how "chronologically compressed" is each group. The coloured rectangles mark the first, second and third proposed epidemic periods. The surge dates identified in each country, and the period they were assigned to, are available in the S1 Fig.

This shows how the distribution of the first group of epidemic surges fits well with the first period proposed, with the majority of surge dates concentrated in the first quarter of 2020. The second and third groups of surges have a moderate temporal overlap, which is greater between the third and fourth, and becomes very pronounced entering 2021.

Thus, the first epidemic surge date detected in Europe (24/02/2020) and the 5th percentile of the second surge (07/05/2020) were used as delimiters of the first period. The latter date is also the point from which the surges are attributed to the second period, until reaching the median of the fourth group of surges (13/02/2021) (out of eight in total). The third period is defined as from this date until the end of the study period (23/11/2021).

### Regression models for epidemic surges

**Linear regression.**   Upon preliminary inspection of the data, we decided not to include the variable for the biweekly average of the proportion of tests carried out during the first wave, since 62 per cent of the records in the first wave had an average of 0, and the remaining cases created an unrealistic association in the regression model.

The full results can be found in Table 1. Neither latitude nor longitude had significant associations with epidemic surge timing in both second and third periods. The biweekly average for the stringency index was found to be negatively associated with the outcome variable in second and third periods. Vaccination was significantly associated with surge timing in the third period, with a delay of 2.25 days per percentage point increase.

**Cox regression.**   The results of this analysis can be seen in Table 2. For similar reasons to those noted in the previous section, the variable for the proportion of tests carried out was not included in the first period. No statistically significant association was found in the first period. In the second period, we found a significant association only with the average of the restriction stringency index, with an increase of the risk of bringing forward the epidemic surge. In the

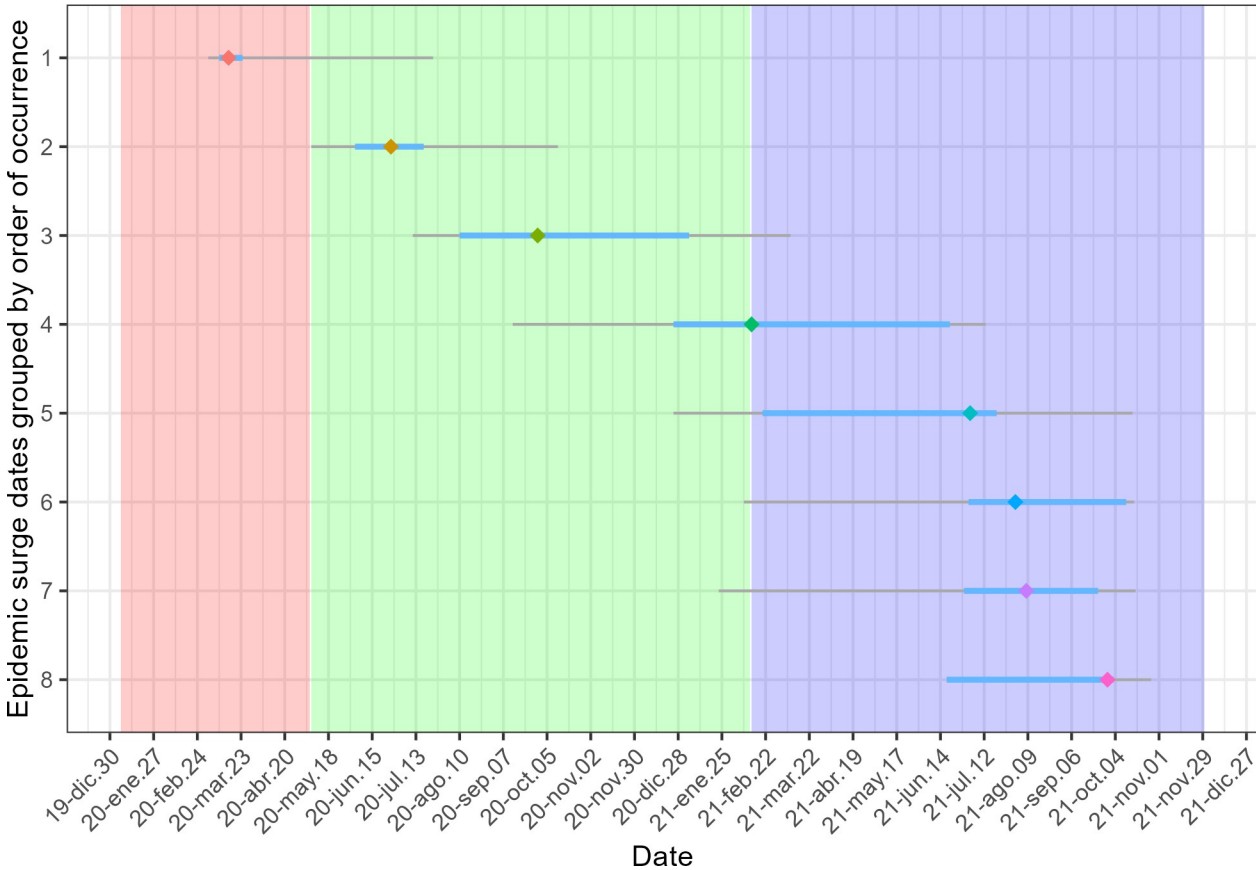

**Fig 2. Temporal distribution of epidemic surge dates, grouped by order of occurrence in each country.** Median dates are showcased as diamonds, interquartile range is in blue and 5th/95th percentiles are in grey color. Red, green and blue rectangles in the background correspond to first (initial-lockdown phase), second (pre-vaccination) and third (vaccine availability) periods. Data as of 23/11/2021.

third period, an association was found with both the stringency index and the completed vaccination percentage, the first increasing the risk and the second as a delaying factor.

## Epidemic intensity models

The results of the linear variables introduced into the GAM models can be seen in Table 3. The results are shown based on the dependent variable studied. In the case of Rt, all the parameters introduced in the model seem to have a statistically significant effect, except for the number of tests per 1000 inhabitants. By the same token, all the variables showed a negative association with Rt. In the models with cumulative incidence and new cases, only the percentage of people vaccinated, latitude and tests per 1000 inhabitants showed a statistically significant association, with a negative association in the first two and a positive in the case of tests. In the three models, latitude has been the variable that has shown the greatest effect on the dependent variable. The random effect of countries, as well as country-specific effects of vaccination and restrictive measures, have been found to be significant in all models.

## Discussion

Our analysis did not show clear associations between geographical location and the timing of epidemic spread throughout Europe. A west-to-east trend did seem to be present during the

**Table 1. Results of the linear regression model.**

| Variables | Coefficient | 95% CI | | p value |
|---|---|---|---|---|
| **First period** | | | | |
| **(Intercept)** | 63.04 | 46.56 | 79.52 | 0.00 |
| **Latitude** | 0.04 | -0.26 | 0.35 | 0.78 |
| **Longitude** | 0.22 | 0.02 | 0.42 | **0.03** |
| **Biweekly stringency index average** | 0.23 | 0.11 | 0.36 | **0.00** |
| **Second period** | | | | |
| **(Intercept)** | 176.60 | 101.94 | 251.26 | 0.00 |
| **Latitude** | 0.59 | -0.77 | 1.96 | 0.38 |
| **Longitude** | -0.44 | -1.30 | 0.42 | 0.30 |
| **Biweekly average of tests/1000 people** | 10.26 | -5.97 | 26.49 | 0.21 |
| **Biweekly stringency index average** | -0.50 | -1.05 | 0.04 | 0.07 |
| **Third period** | | | | |
| **(Intercept)** | 536.23 | 443.05 | 629.41 | 0.00 |
| **Latitude** | -0.19 | -1.71 | 1.32 | 0.80 |
| **Longitude** | 0.58 | -0.47 | 1.62 | 0.27 |
| **Biweekly average of tests/1000 people** | 0.01 | -1.35 | 1.36 | 0.99 |
| **Biweekly stringency index average** | -1.36 | -2.17 | -0.55 | **0.00** |
| **% of population vaccinated at 14 days before** | 2.25 | 1.33 | 3.17 | **0.00** |

Association of the variables studied with the moment of epidemic surge in the three epidemic periods.

first epidemic period (start of the pandemic until the end of the initial generalized lockdowns). Both the stringency index, measuring the severity of NPIs, and the vaccination coverage seemed to have some predictive value for the timing of epidemic surges. As for the three analyzed measures of epidemic intensity (reproduction number, cumulative incidence, new disease cases), we found a consistent north-to-south gradient. Vaccination coverage was the only

**Table 2. Results of the Cox regression model.**

| Variable | OR | 95% CI | | p.value |
|---|---|---|---|---|
| **First period** | | | | |
| **Latitude** | 0.98 | 0.94 | 1.03 | 0.47 |
| **Longitude** | 1.00 | 0.97 | 1.03 | 0.88 |
| **Biweekly stringency index average** | 1.01 | 0.99 | 1.03 | 0.41 |
| **Second period** | | | | |
| **Latitude** | 0.99 | 0.95 | 0.99 | 0.56 |
| **Longitude** | 1.02 | 0.99 | 1.02 | 0.15 |
| **Biweekly stringency index average** | 1.03 | 1.01 | 1.03 | **0.01** |
| **Biweekly average of tests/1000 people** | 0.79 | 0.49 | 0.79 | 0.35 |
| **Third period** | | | | |
| **Latitude** | 1.00 | 0.95 | 1.00 | 0.99 |
| **Longitude** | 0.99 | 0.96 | 0.99 | 0.55 |
| **Biweekly stringency index average** | 1.05 | 1.02 | 1.05 | **0.00** |
| **Biweekly average of tests/1000 people** | 1.00 | 0.97 | 1.00 | 0.92 |
| **% of population vaccinated at 14 days before** | 0.96 | 0.93 | 0.96 | **0.02** |

Association of the variables studied with the moment of epidemic surge in the three epidemic periods.

**Table 3. Results of the generalized additive model.**

| Variable | Coefficient | 95%CI | | p value |
|---|---|---|---|---|
| **Rt:** | | | | |
| (Intercept) | 3.254210 | 1.942628 | 4.565792 | 0.000001 |
| Stringency index | -0.007375 | -0.010211 | -0.004539 | **0.000000** |
| % of people fully vaccinated | -0.009167 | -0.011648 | -0.006687 | **0.000000** |
| Latitude | -0.027960 | -0.053695 | -0.002225 | **0.033243** |
| Longitude | -0.017321 | -0.033486 | -0.001157 | **0.035732** |
| Tests per thousand inhabitants | -0.000826 | -0.001925 | 0.000272 | 0.140241 |
| **Cumulative incidence:** | | | | |
| (Intercept) | 3,678.269552 | 1,641.590729 | 5,714.948375 | 0.000402 |
| Stringency index | 2.945669 | -2.088341 | 7.979678 | 0.251456 |
| % of people fully vaccinated | -10.492031 | -14.848994 | -6.135068 | **0.000002** |
| Latitude | -64.414226 | -104.398486 | -24.429965 | **0.001596** |
| Longitude | -14.563171 | -39.738007 | 10.611665 | 0.256902 |
| Tests per thousand inhabitants | 17.351197 | 16.139947 | 18.562446 | **0.000000** |
| **New cases per million inhabitants:** | | | | |
| (Intercept) | 2,558.057186 | 1,208.365359 | 3,907.749013 | 0.000204 |
| Stringency index | 0.475945 | -2.797642 | 3.749532 | 0.775682 |
| % of people fully vaccinated | -7.823144 | -10.657514 | -4.988775 | **0.000000** |
| Latitude | -43.110664 | -69.607772 | -16.613556 | **0.001432** |
| Longitude | -10.085037 | -26.768503 | 6.598428 | 0.236128 |
| Tests per thousand inhabitants | 11.525056 | 10.717544 | 12.332569 | **0.000000** |

Association of the linear variables introduced with the three epidemic intensity measures.

additional variable showing a significant association with epidemic intensity. These findings suggest the importance of socioeconomic, cultural and other heterogeneities in the dynamic behaviour of epidemic spread across countries.

## Epidemic surges and periods

Examples of precise definitions of epidemic periods are scarce in the literature; these are difficult events to define at an international level. Our epidemic period definitions are based merely on guidelines considering the differential characteristics of each (the moment of the epidemic or the presence of vaccines), although the analysis shown in Fig 2 demonstrates that it is possible to distinguish with some precision the first period from the second. The separation between the second and third periods is more complicated, as a greater degree of temporal overlap with the surge groups is evident, which increases during this last period. One reason why this phenomenon may occur is that, as vaccination began in Europe, the control of transmissibility at specific times was achieved more quickly, and with less severe non-pharmacological measures, which favours more frequent epidemic surges.

**Regression models for epidemic surges.** Our results for the chronology of the first pandemic period in Europe are heterogeneous. None of the variables that are statistically significant in the linear model are significant in the Cox model.

The linear model suggests a possible west-east movement, which would fit well with the regions where the virus initially manifested itself in Europe. Likewise, the stringency index shows a positive association with the delay in the moments of epidemic surge. Therefore, it seems reasonable to think that in those countries which had not previously experienced any

other epidemic surge, the delay would be more closely linked to the level of restrictions imposed.

Regarding the second and third periods, in both models the longitude seems to show a stronger association (although with p > 0.05) than latitude, which could suggest a longitudinal axis in the movement of the pandemic through Europe.

Likewise, we found an inverse relationship between the biweekly average of the stringency index and the delay in epidemic surges in the second and third waves, which is verified by both models. This could be because the stringency index may be acting as a proxy for other variables not considered in the models: for example, a high stringency index score is likely to be associated with prior high CI figures, which in turn pave the way for a potential rebound in transmissibility by increasing the rate of effective contacts. In these periods, the vaccination percentage variable is statistically significant in both models. In the linear model, we found a delay of 2.25 days for each percentage point of full vaccination.

Analysis of the pandemic's temporal development across Europe in the literature are scarce. Among these, Walrand [6] also studies epidemic surge dates, however calculated differently from ours. Its conclusions point to a latitudinal gradient that manifests itself in the second epidemic period, and which relates to low population 25-OH-vitamin D levels, placing the first surges of the period in countries at high latitudes. As already mentioned, our results indicate, in any case, a preponderance of the longitudinal axis in terms of dispersion. In the case of other respiratory diseases such as influenza, research has been carried out with a similar methodology to approximate dispersion directionality across Europe [9], although the weeks with maximum clinical activity were used as a temporal variable. The results of that study also seem to point towards greater importance for the longitudinal axis over the latitudinal in the European context.

However, there is evidence supporting that local dynamics and well-connected urban areas are the most resilient/persistent pathways for the virus to spread, and on the contrary, great/ large scale spreading wanes rapidly [10]. Indeed, our findings suggest that the pandemic did not take the form of a sweeping wave across Europe, at least as a solid, recurrent pattern.

## Epidemic intensity models

For the three measures of epidemic intensity evaluated, it is latitude that offers the most conclusive results, although the longitude is also significant when Rt is analysed.

The environmental characteristics associated with each geographical position may be responsible for these patterns. Factors such as humidity (absolute and relative), average temperature and solar radiation have been studied to explain seasonality and geographical differences in the incidence of some respiratory infectious diseases, including those caused by other coronaviruses, especially in areas of subtropical latitude or greater [11–14]. Furthermore, several studies describe a relationship between latitude and incidence and mortality figures for SARS-CoV-2 [15], but this evidence points in the opposite direction to that of our results (although there are important methodological differences with our study: the results were based on a correlation analysis without additional explanatory variables), attributing this gradient to temperature, which is indicated as a preventative factor–with an inverse correlation having been found in other studies [7, 16, 17].

Our results may not exclusively reflect differences caused by variations in environmental conditions. The cultural and socio-economic differences between the countries of northern and southern Europe are already well known. Temperature itself can condition social behaviours that favour the virus's transmissibility [18].

The only statistically significant effect of the stringency index appears to be on transmissibility, in a negative sense. Given that the most severe restrictions have been imposed due to rises in CI or daily new cases, it is logical that we have not found a negative association with these variables.

According to our results, the effect of mobility restrictions implies a maximum average reduction (taking into account that the restriction stringency index is constructed from 0 to 100) of 0.7 points in the Rt. This is a rather low figure compared to other estimates of the effect of non-pharmacological interventions on the Rt, given at up to an 81% reduction in the case of home confinement [19], which could achieve a 1.2 points reduction, given a 1.5 Rt. Other studies using the OxCGRT describe inflexion points in the stringency index that are required to achieve certain trend changes in the effective reproduction number (a minimum of 79.6 in the index to reduce the Rt below 1 in Europe) [20].

The respective coefficients for vaccinated population percentage show a negative association with all intensity measures used; it is associated with a reduction in transmissibility of 11% more than that of the restrictive measures. Studies from Italy [21] and Israel [22] also attribute reductions in infections to their vaccination campaigns.

Finally, the effect by country and country-specific effects on vaccination and restrictive measures suggest additional differences between countries, paving the way for further research on the nature of these effects [14]. We believe that these differentiating aspects may play an important role in the transmission dynamics of COVID-19.

## Strengths and limitations

This study contributes to the scarcely treated matter of the temporal development of the pandemic in Europe, giving clues as to possible axes of the virus's spread through the continent. We used extensive data from 43 countries and multiple analytical methods and epidemiological variables. The results raise questions about the role of environmental and sociocultural variables as predictors of the magnitude of the epidemic, and the need to disentangle them from the effect of geographic location. Apparent contradictions with other studies might not be as they seem, as this relationship between geographical position and COVID epidemiological variables has not been thoroughly studied in our opinion, and thus, is not well established.

This study has a number of limitations, that may have prevented us from obtaining more precise and statistically detailed conclusions. The need to establish clear and mutually exclusive divisions between the different epidemic periods in Europe (based on the epidemic periods themselves) introduces a distortion in the analysis of those very same periods, since it is an artificial discontinuity that in turn has conditioned the selection of the epidemic surge dates for each country. We initially introduced vaccination percentage, restriction stringency index, testing, latitude and longitude in epidemic intensity models as non-linear, and the results showed that the relationship was, in fact, linear. The linearity assumption could be a limiting factor in the models explaining surge dates, as non-linear associations could be due to heterogeneous features across countries.

In addition, the unit of analysis has been limited to areas corresponding to countries, with their associated centroids. Had we been able to use smaller units, centroids could then have better represented the geographical position of each. Also, disaggregated data would have provided greater statistical power to our analysis. In addition, latitude and longitude reflect, beyond climatic differences, political and social differences, complicating their interpretation. Finally, other adjustment variables, such as population density, pollution levels and people's movements, have not been included in the analyses.

## Conclusions

This paper shows the importance of European countries' latitude in explaining the differences in the intensity with which the pandemic has manifested itself in each; however, we cannot conclude that these differences are due to environmental variables, which leads us to point out the potential role of other differentiating aspects between countries. There are important social, political and economic differences between southern and northern Europe that might explain our results. Further investigation is needed to identify which factors are relevant to this matter, and how they would overcome environmental drivers, if they truly influence COVID-19 epidemiological behaviour as reported in literature [15]. Although local and regional dynamics have been identified as useful predictors of long-term behaviour of the virus at that same scale [10], there is still uncertainty as to whether the timing of large scale epidemic events can be influenced by other factors.

Restrictive measures affect the transmissibility of SARS-CoV-2, although they do not seem to delay epidemic surges. As for the vaccination rate, it seems to play a role in this delay, as well as reducing transmissibility and other epidemic intensity measures. A longitudinal axis of dispersion for the pandemic in Europe seems plausible, although more evidence is still needed.

## Supporting information

**S1 Fig. Identified surge dates for every country in the study.** Colored rectangles in the background mark the proposed epidemic periods. Surge dates are colored accordingly to show the periods they were assigned to.
(PDF)

## Acknowledgments

We are grateful to J. Segú-Tell for the data cleaning support.

## Author Contributions

**Conceptualization:** Diana Gómez-Barroso.

**Data curation:** Alejandro Martínez-Portillo.

**Formal analysis:** Alejandro Martínez-Portillo.

**Methodology:** Alejandro Martínez-Portillo, David Garcia-Garcia, Diana Gómez-Barroso.

**Software:** Alejandro Martínez-Portillo, David Garcia-Garcia.

**Supervision:** Diana Gómez-Barroso.

**Validation:** David Garcia-Garcia.

**Visualization:** Alejandro Martínez-Portillo.

**Writing – original draft:** Alejandro Martínez-Portillo.

**Writing – review & editing:** David Garcia-Garcia, Inmaculada Leon, Rebeca Ramis-Prieto, Diana Gómez-Barroso.

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
