## [Decision Letter · Decision Letter 0]

15 May 2023

PONE-D-22-29647Latitude and longitude as drivers of COVID-19 waves’ behavior in Europe: a time-space perspective of the pandemicPLOS ONE

Dear Dr. Martínez-Portillo,

Thank you for submitting your manuscript to PLOS ONE. After careful consideration, we feel that it has merit but does not fully meet PLOS ONE’s publication criteria as it currently stands. Therefore, we invite you to submit a revised version of the manuscript that addresses the points raised during the review process.

Both reviewers have raised major concerns on the original contribution of the paper. The editor agrees with the two reviewers on this key point that needs to be carefully addressed. Please make major effort in discussing the contributions clearly and appropriately, amid other major/minor comments provided by reviewers.

We look forward to receiving your revised manuscript.

Kind regards,

Chenfeng Xiong

Academic Editor

PLOS ONE

2. We note that Figure 1 in your submission contain [map/satellite] images which may be copyrighted. All PLOS content is published under the Creative Commons Attribution License (CC BY 4.0), which means that the manuscript, images, and Supporting Information files will be freely available online, and any third party is permitted to access, download, copy, distribute, and use these materials in any way, even commercially, with proper attribution. For these reasons, we cannot publish previously copyrighted maps or satellite images created using proprietary data, such as Google software (Google Maps, Street View, and Earth). For more information, see our copyright guidelines: http://journals.plos.org/plosone/s/licenses-and-copyright.

Reviewers' comments:

Reviewer's Responses to Questions

**Comments to the Author**

1. Is the manuscript technically sound, and do the data support the conclusions?

Reviewer #1: Partly

Reviewer #2: Partly

2. Has the statistical analysis been performed appropriately and rigorously? 

Reviewer #1: Yes

Reviewer #2: Yes

3. Have the authors made all data underlying the findings in their manuscript fully available?

Reviewer #1: Yes

Reviewer #2: Yes

4. Is the manuscript presented in an intelligible fashion and written in standard English?

Reviewer #1: Yes

Reviewer #2: No

5. Review Comments to the Author

Reviewer #1: The study investigates the spatial gradient of COVID spread in Europe along with the influences of vaccination and NPI practice. The paper is well organized with method description and result discussion but the contributions are not so clear and thus seem limited. How can the pandemic spread direction (if proved) help? Also, since the linear regression and COX regression yield significant results for different periods and in the third period, suggest different relations between epidemic surge data and stringency index/vaccinated percentage, how can we believe they generated convincing indications? There is a great summary of limitations listed but there lacks discussion on whether those limitations would distort the model results and conclusions.

Minor comments:

1. How is the grouping of surge dates performed in Fig 2? Why eight groups? There are some definition on line 133-134 but it’s confusing. On line 138, there is no clear definition of periods but only vague ranges. The related results are confused too (line 197-200). What do they indicate?

2. There is a mixed use of stringency index and restriction index. Suggest a consistent term if they mean the same thing. Similar comment for first period and first wave.

3. Suggest a better way of mentioning the surge dates in each county (line 188-190) in appendix, e.g., with a general summary of the trends.

4. Line 222-225, negative relationship for stringency index and positive one for vaccination. Line 248-249, positive relationship for stringency index and negative one for vaccination. Contradictory results? Related concern in Line 307-309, with the current data, it seems feasible to evaluate the assumption.

5. Line 284, figure x?

6. Line 334, opposite from existing study, who to believe?

7. As stated, the environmental and sociocultural variables are not considered, how helpful is the trend by latitude and longitude?

Reviewer #2: I read the paper entitled on the" Latitude and longitude as drivers of COVID-19 waves’ behavior in Europe: a time-space perspective of the pandemic". I have few major concerns regarding the work. My comments are as follows:

• The novelty of the work should be conceptualized here in the introduction.

• The map of selected European countries should be regenerated in some map Making software e.g. Arc Map, GeoDa etc. instead of open street map.

• The reason for using different regression models must also be mentioned clearly in the methodological framework.

• It seems that more explanations about methods need to be included. Why they are superior in this case?

• Is there any special reason to use GAM for analysis?

• The results should be very concise focusing on the title of the paper. It should be arranged properly.

• The discussion should be in relevant with the existing studies on the subject matter.

• The conclusions section is not enough. It should be strengthened.

6. PLOS authors have the option to publish the peer review history of their article (what does this mean?). If published, this will include your full peer review and any attached files.

Reviewer #1: No

Reviewer #2: **Yes: **Munazza Fatima

---

## [Author Response · Author response to Decision Letter 0]

3 Aug 2023

First, we wanted to apologize for an error we detected through the revision process. Originally we stated that the analysis was performed over 44 countries, but there was no data available for Vatican City and it was automatically excluded from the analysis. We never changed the country list in the initial submission, but it is corrected in this version, so there are 43 countries in total. Thank you for your time. 

Reviewer #1: The study investigates the spatial gradient of COVID spread in Europe along with the influences of vaccination and NPI practice. The paper is well organized with method description and result discussion but the contributions are not so clear and thus seem limited. How can the pandemic spread direction (if proved) help? Also, since the linear regression and COX regression yield significant results for different periods and in the third period, suggest different relations between epidemic surge data and stringency index/vaccinated percentage, how can we believe they generated convincing indications? There is a great summary of limitations listed but there lacks discussion on whether those limitations would distort the model results and conclusions.

As we conclude in our study, a determined spread direction for every pandemic period does not seem plausible. It was not completely discarded during the first period, but our results are not consistent. In any case, we think our hypothesis was worth checking, since any knowledge about when an epidemic is expected to hit a country can be an important asset to improve preparedness for possible future epidemic events. 

Although Cox and linear regression show some differences in the first period analyzed, they do not differ on the relations between epidemic surge data and stringency index or vaccinated percentage. Note that the models’ establish the same associations with opposing signs. Negative coefficients in the linear model imply earlier surge dates, while positive, greater than 1 hazard ratios in Cox model suppose an increased risk of surging. We have included a clarification in the corresponding section (see lines 173-175).

For these reasons, we do believe the contributions of our work are sound and useful. First, we present evidence towards the lack of a significant, consistent spread direction of the disease, and second, we analyze the effect of the variables considered in the study in a key dependent variable that has not been addressed in detail in the literature: the surge dates of epidemic waves. We do so by considering data on several countries in Europe in our analysis and several methods, features that we believe make a strong case for the relevance of our findings.

Thank you for pointing out the room for improvement on the limitations section. We tried to make this section clearer, pointing out to some specific aspects that could have been improved. We think we are now sufficiently transparent on the shortcomings of our study, but the lack of precedent articles on the topic complicate commenting further on the possible distortions introduced by our methods. 

Minor comments:

1. How is the grouping of surge dates performed in Fig 2? Why eight groups? There are some definition on line 133-134 but it’s confusing. On line 138, there is no clear definition of periods but only vague ranges. The related results are confused too (line 197-200). What do they indicate?

Surge dates grouping is carried out as follows:

- Surge dates for every country are obtained, using the epidemic surge definition (Rt>1 for as long as 14 days or more). The start of those time lapses (day 1) is selected as surge date.

- Dates are ordered chronologically (date nº1, date nº2…)

- Every date is grouped by number of order. 

- The maximum number of surges observed at a single country was eight, so there was a total of eight groups.

We have updated the methods and results sections to be more clear about this process, and to explicitly mention the particular choices we assume for this procedure.

2. There is a mixed use of stringency index and restriction index. Suggest a consistent term if they mean the same thing. Similar comment for first period and first wave.

They mean the same thing and the text has been corrected. 

3. Suggest a better way of mentioning the surge dates in each county (line 188-190) in appendix, e.g., with a general summary of the trends.

We elaborated a new plot which features the evolution of the Rt and the surge dates obtained for every country, marking the period to which they were assigned. 

4. Line 222-225, negative relationship for stringency index and positive one for vaccination. Line 248-249, positive relationship for stringency index and negative one for vaccination. Contradictory results? Related concern in Line 307-309, with the current data, it seems feasible to evaluate the assumption.

These are not contradictory results. Similar associations are obtained with both methods, as is described above. We adressed this issue in the text, trying to avoid any misleading wordings. 

5. Line 284, figure x?

Corrected. Figure 2. 

6. Line 334, opposite from existing study, who to believe?

The study authored by Burra et al. describes a negative relationship between temperature/latitude and COVID-19 epidemiological variables, and this is indeed an opposite finding to that of our study, but these results apply to worldwide data. When the same analysis is performed with US data only, latitude and temperature are no longer associated with incidence or mortality. In addition, their analysis is limited to correlation, not regression, and thus, their results are not adjusted by relevant covariables such as implemented restrictions. 

Our analysis is also restricted to a limited range of latitude, which could be insufficient to fully uncover the effect geographical position had on epidemiological variables. But, as is discussed in the paper, we believe latitude in Europe is correlated with important features at country level, beyond geographical/meteorological differences. These features would overcome the potencial effects of temperature, which is the main environmental variable analysed in literature. In other words, it may be that our results are determined not by the physical environment on which Europe sits, but other features that are unique to the continent. Apparent contradictions with other studies might not be as they seem, as this relationship between geographical position and COVID epidemiological variables has not been thoroughly studied in our opinion, and thus, is not well established. We have updated the discussion to further comment on this matter.

7. As stated, the environmental and sociocultural variables are not considered, how helpful is the trend by latitude and longitude?

We did not expect to find such contrasting results with other studies. When we introduced latitude and longitude in the models, we intended to use them as proxys of environmental variables. 

In the light of our results, it becomes clear that latitude or longitude are insufficient to explain the differences between European countries. We have updated the discussion with this specific point. Still, for the reasons stated above, we think that the apparent contradictions are valuable and give way to further investigation. 

Reviewer #2: I read the paper entitled on the" Latitude and longitude as drivers of COVID-19 waves’ behavior in Europe: a time-space perspective of the pandemic". I have few major concerns regarding the work. My comments are as follows:

• The novelty of the work should be conceptualized here in the introduction.

We have added a more detailed explanation.

• The map of selected European countries should be regenerated in some map Making software e.g. Arc Map, GeoDa etc. instead of open street map.

Done, thank you.

• The reason for using different regression models must also be mentioned clearly in the methodological framework.

We have updated the methods section to explicitly state our reasoning in the use of the three complementary approaches.

• It seems that more explanations about methods need to be included. Why they are superior in this case?

We included more detailed reasoning on the strengths and purpose of each of the methodological approaches we utilize in our work. The methods we used are well-established and extensively employed in the literature and, since we faced an unusual type of analysis, and considering it was scarcely treated in the literature, we leaned towards this comprehensive approach, allowing for extensive investigation and comparison, using the known methods that could best serve our purposes. 

• Is there any special reason to use GAM for analysis?

We wished to account for time dependency in our analysis, and generalized additive models are a versatile tool to include linear and non-linear dependencies of parameters such as time, or dynamic variables that evolve over time (which is not possible within linear or Cox regression models). 

• The results should be very concise focusing on the title of the paper. It should be arranged properly.

Corrected; thank you for the remark.

• The discussion should be in relevant with the existing studies on the subject matter.

We included additional studies to update the discussion to current knowledge.

• The conclusions section is not enough. It should be strengthened.

We have included more details and relevant comments in the section.

---

## [Decision Letter · Decision Letter 1]

4 Sep 2023

Latitude and longitude as drivers of COVID-19 waves’ behavior in Europe: a time-space perspective of the pandemic

PONE-D-22-29647R1

Dear Dr. Martínez-Portillo,

We’re pleased to inform you that your manuscript has been judged scientifically suitable for publication and will be formally accepted for publication once it meets all outstanding technical requirements.

Kind regards,

Chenfeng Xiong

Academic Editor

PLOS ONE

Additional Editor Comments (optional):

Reviewers' comments:

Reviewer's Responses to Questions

**Comments to the Author**

1. If the authors have adequately addressed your comments raised in a previous round of review and you feel that this manuscript is now acceptable for publication, you may indicate that here to bypass the “Comments to the Author” section, enter your conflict of interest statement in the “Confidential to Editor” section, and submit your "Accept" recommendation.

Reviewer #1: All comments have been addressed

Reviewer #2: All comments have been addressed

2. Is the manuscript technically sound, and do the data support the conclusions?

Reviewer #1: Yes

Reviewer #2: Yes

3. Has the statistical analysis been performed appropriately and rigorously? 

Reviewer #1: Yes

Reviewer #2: Yes

4. Have the authors made all data underlying the findings in their manuscript fully available?

Reviewer #1: Yes

Reviewer #2: Yes

5. Is the manuscript presented in an intelligible fashion and written in standard English?

Reviewer #1: Yes

Reviewer #2: Yes

6. Review Comments to the Author

Reviewer #1: (No Response)

Reviewer #2: (No Response)

7. PLOS authors have the option to publish the peer review history of their article (what does this mean?). If published, this will include your full peer review and any attached files.

Reviewer #1: No

Reviewer #2: **Yes: **Munazza Fatima

---

## [Editor Report · Acceptance letter]

7 Sep 2023

PONE-D-22-29647R1 

Latitude and longitude as drivers of COVID-19 waves’ behavior in Europe: a time-space perspective of the pandemic 

Dear Dr. Martínez-Portillo:

I'm pleased to inform you that your manuscript has been deemed suitable for publication in PLOS ONE. Congratulations! Your manuscript is now with our production department. 

Kind regards, 

on behalf of

Dr. Chenfeng Xiong 

Academic Editor

PLOS ONE